# Molecular characterization of *Blastocystis* subtypes in symptomatic patients from the southern region of Syria

**Buthaina Darwish, Ghalia Aboualchamat◉, Samar Al Nahhas◉***

Department of Animal Biology, Faculty of Science, Damascus University, Damascus, Syria

* samar.nahhas@yahoo.com, samar.nahhas@damascusuniversity.edu.sy

**Data Availability Statement:** All relevant data are within the manuscript.

## Abstract

*Blastocystis* sp. is an enteric protist found in humans and a wide range of animal hosts. Genetic variations were established among the 38 different subtypes detected so far, 14 of which are commonly found in human and animal hosts. The aim of the present study is to estimate the prevalence of the common *Blastocystis* subtypes and evaluate the possible correlation with several variables (gender, age, symptoms, domestic animals. . .), among patients from the southern region of Syria. Fecal samples were collected from individuals suffering from gastrointestinal complaints. Microscopic examination along with genotype analyses using seven pairs of subtype-specific primers was performed. Our results revealed the presence of *Blastocystis* sp. in 46 isolates out of the 60 samples microscopically studied (76.7%); single infection was detected in 24 isolates whereas co-infection with other protozoa was identified in 22 ones. Molecular detection targeting the *SSU rRNA* gene revealed a 100% positive presence of *Blastocystis* sp. in all the samples. Genotyping results detected the presence of five different subtypes (ST1-ST5) with varying proportions. However, ST1 was the dominant subtype observed (66.7%). Mixed subtype infections were found in 9 isolates (15%). Three samples remained undefined, nonetheless. Our statistical results showed no significant correlation between *Blastocystis* STs infection and the different studied variables. In conclusion, this study provides the first genetic characterization of *Blastocystis* subtypes prevalence in patients from the southern region of Syria. ST1 distribution was highly predominant. Further molecular studies are needed to estimate the prevalence of *Blastocystis* sp. infection in other regions in Syria and to understand the epidemiology and sources of transmission to humans.

## Introduction

*Blastocystis* sp. is a gastrointestinal protozoan found in humans and many animals [1, 2]. It has a large distribution worldwide, with increasing cases in developing countries, due to poor hygiene practices and consumption of contaminated food or water [3, 4]. The pathogenicity of this parasite is questionable. For example, according to some studies, *Blastocystis* has no clinical relevance, while recent associations have been found between *its* presence and some

**Funding:** the authors received no specific funding for this work.

**Competing interests:** The authors have declared that no competing interests exist.

specific gastrointestinal symptoms [5, 6]. An extensive genetic diversity has been observed among numerous *Blastocystis* sp. isolated from different hosts [7, 8]. At present, 38 subtypes (STs) have been identified (ST1-ST38) [9–11], 14 of which have been isolated from both humans and animals worldwide [12–14], while the other STs have been reportedly found only in animals [15, 16]. Researchers have found that STs 1–4 appear to be the most common *Blastocystis* inhabitants of the human intestines, and the other 10 STs are rare in humans but are frequently detected in various animal groups including birds and hoofed animals [1, 16, 17]. Molecular identification studies of *Blastocystis* STs provide a discriminating tool for investigating the epidemiology of the parasite including transmission route, host specificity, and chemotherapeutic drug resistance [18, 19]. In Syria, *Blastocystis* sp. is not well studied and there is a lack of information on its prevalence, distribution, and its diverse subtypes. Therefore, the present study aimed to estimate the infection incidence of the different common subtypes among resident patients in the southern region of Syria and their possible correlation with several variables.

## Materials and methods

### Ethics approval and informed consent

Damascus University approved the aims and the procedures of the study (Ref. No: 4031/2019). Written informed consent was obtained from all patients participating in the study, or their parents or legal guardians.

### Sampling and the studied specimen

A cross-sectional study was conducted between the period of November 2019 and March 2020. Individuals attending outpatient clinics at Al-Assad University Hospital, the Syrian Specialty Hospital and Dara Health Center, suffering from various gastrointestinal disorders, were recruited in this study. All patients were residents from the southern region of Syria (including the city of Damascus and its countryside, Dara' governorate and its countryside). General information was obtained from each patient including age, gender, general health conditions, source of drinking water, contact with domestic animals like cats and dogs as well as symptoms such as diarrhea, abdominal spam, flatulence, anorexia/weight loss, . . .etc. Stool specimens were collected in clean sterile plastic containers. Each specimen was divided into two parts: one part was fixed in formalin solution 10% (1:3) for microscopic investigation and the other was stored at -20˚C for molecular studies.

### Microscopic examination

Approximately 1 mg of each fixed sample was stained by Lugol's iodine as described in [20] and examined directly using light microscope (×40 magnifications). The diagnostic criterion adopted in determining the positivity of *Blastocystis sp*. infection was the detection of at least 5 vacuolar forms [21].

### Molecular detection and subtyping

The total genomic DNA was extracted from almost 250–300 mg of each fecal sample collected using QIAamp-DNA stool mini kit (QIAamp-DNA Stool Mini Kit, QIAGEN) as described in [22] and stored at -20˚C until use.

The small subunit ribosomal RNA (*SSU-rRNA*) gene was used for the detection of *Blastocystis* sp. in all the studied samples. For each specimen, 4 μl of the extracted template DNA was amplified using a pair of specific diagnostic primers (b11400 FORC and b11710. REVC) [23].

**Table 1. The primer sets used for genotyping.**

| STS primers sets / subtype | Sequences of forward and reverse primers | Product size |
|---|---|---|
| SB 83 / ST1 | F: GAAGGACTCTCTGACGATGA | 351 (bp) |
| | R: GTCCAAATGAAAGGCAGC | |
| SB340 / ST2 | F: TGTTCTTGTGTCTTCTCAGCTC | 704 (bp) |
| | R: TTCTTTCACACTCCCGTCAT | |
| SB227 / ST3 | F: TAGGATTTGGTGTTTGGAGA | 526 (bp) |
| | R: TTAGAAGTGAAGGAGATGGAAG | |
| SB337 / ST4 | F: GTCTTTCCCTGTCTATTCTGCA | 487 (bp) |
| | R: AATTCGGTCTGCTTCTTCTG | |
| SB336 / ST5 | F: GTGGGTAGAGGAAGGAAAACA | 317 (bp) |
| | R: AGAACAAGTCGATGAAGTGAGAT | |
| SB332 / ST6 | F: GCATCCAGACTACTATCAACATT | 338 (bp) |
| | R: CCATTTTCAGACAACCACTTA | |
| SB155 / ST7 | F: ATCAGCCTACAATCTCCTC | 650 (bp) |
| | R: ATCGCCACTTCTCCAAT | |

Each PCR reaction was conducted in 25µl final volume consisting of 12.5 µl One PCR™ master mix 2X (GeneDirex Inc, Taiwan ROC), 1 µl of each primer, and 10.5 µl nuclease-free water. Amplification conditions consisted of initial denaturation at 94˚C for 3 min, 30 cycles including denaturation at 94˚C for 1 min, annealing at 58˚C for 1 min and extension at 72˚C for 1 min. The final extension was at 72˚C for 5 minutes.

Seven subtype-specific sequence-tagged-site (STS) primers were chosen for the identification of the different STs studied as shown in Table 1 [24]. The PCR cycling conditions were as described by [25].

All PCR experiments contained a negative control (4 µl of nuclease-free water) for contamination detection. PCR reactions were carried out using Eppendorf Master Cycler. The PCR products were electrophoresed in 1.5–2% agarose gel stained with ethidium bromide (Sigma-Aldrich, USA) along with a 100 bp DNA ladder (GeneDirex Inc, Taiwan ROC) as a standard size.

## Statistical analysis

All data were analyzed using IBM SPSS Statistics Version 25.0 (SPSS, Inc.; Chicago, IL, USA). Percentages were used to describe the prevalence of the different Blastocystis STs. Pearson's chi-squared and Fisher's Exact tests were used to assess the possible association between subtypes and the different studied variables. A *P* value of $\leq 0.05$ was considered statistically significant.

## Results

A total of 60 outpatients, who visited hospitals and health centres located in the southern region of Syria with different digestive complaints, participated in this study. The female to male ratio was approximately 1.4 (58.3%; 41.7% respectively). In general, the age of patients ranged from 3 to 76 years. The mean was 31.9 ± 21.3 and the median age was 26 years old.

Microscopic analysis showed that only 46 (76.7%) samples were positive for the presence of vacuolar forms of *Blastocystis* sp., whereas molecular detection revealed the existence of *Blastocystis* sp. in all the samples (n = 60, 100%).

Furthermore, our genotyping results detected the presence of five different STs in 57 isolates (95%), while three samples remained undefined (5%). A single ST infection was found in

**Table 2. The prevalence of *Blastocystis* subtypes with/without other intestinal parasites infections.**

| | DIFFERENT SUBTYPES | | | | | | |
|---|---|---|---|---|---|---|---|
| | ST1 | ST2 | ST3 | ST4 | ST5 | MIX STS | UNKNOWN STS |
| **Single Infection (*Blastocystis* sp.): total (n = 33)** | **18** | **0** | **7** | **0** | **1** | **5** | **2** |
| (%) | **54.5%** | **0** | **21.2%** | **0** | **3.03%** | **15.2%** | **6.1%** |
| **Protozoa Co-Infection: total (n = 27)** | **11** | **3** | **6** | **1** | **1** | **4** | **1** |
| (%) | **40.7%** | **11.1%** | **22.2%** | **3.7%** | **3.7%** | **14.8%** | **3.7%** |
| **Co-infection** | | | | | | | |
| *Blastocystis* sp + *Entamoeba* sp | 7 | 0 | 0 | 1 | 1 | 0 | 1 |
| *Blastocystis* sp + *Entamoeba* sp + *Entamoeba coli* | 0 | 1 | 1 | 0 | 0 | 1 | 0 |
| *Blastocystis* sp + *Entamoeba coli* | 2 | 1 | 5 | 0 | 0 | 2 | 0 |
| *Blastocystis* sp + *Giardia* | 1 | 1 | 0 | 0 | 0 | 0 | 0 |
| *Blastocystis* sp + *Entamoeba* sp + *Giardia* | 1 | 0 | 0 | 0 | 0 | 1 | 0 |

48 samples (84.2%), while mixed STs were found in nine isolates (15.8%) as follows: ST1+ST3 (n = 5), ST1+ST2 (n = 3) and ST1+ST2+ST3 (n = 1).

The infection with different STs alone was detected in more than half of the samples (33, ~55%), while co-infection with other intestinal parasites was found in 27 samples (45%). Interestingly, ST1 infection was predominant in both cases (55%, and 41% respectively) (Table 2).

Furthermore, to exclude intestinal manifestations caused by other parasites, the clinical disorders in patients infected with *Blastocystis* sp. alone (31 isolates) were analyzed. Our results showed that the most frequent symptoms were flatulence, abdominal pain, and abdominal spam (61.3%, 58.1% and 54.8% respectively) (Table 3).

Our statistical analysis results showed no significant correlation between any of the STs detected and between age groups, gender, drinking water supply, contact with domestic animals nor between mechanical vectors, as shown in Table 4.

Additionally, no significant correlation was found between symptoms and any STs detected in our study (Table 5).

## Discussion

*Blastocystis* sp. is an intestinal parasite whose pathogenic role is underestimated; several studies reported its presence in both asymptomatic and symptomatic patients [26, 27]. In Syria, epidemiological and molecular studies on this parasite are scarce. This study is the first to identify

**Table 3. The clinical symptoms relevant to the *Blastocystis* subtypes detected.**

| Clinical symptoms | Different Subtypes | | | | | Total |
|---|---|---|---|---|---|---|
| | | | | | | N = 31 |
| | ST1 | ST3 | ST5 | ST1+ST2 | ST1+ST3 | (%) |
| | (n = 18) | (n = 7) | (n = 1) | (n = 3) | (n = 2) | |
| Abdominal pain | 11 | 4 | 1 | 2 | - | 18 (58.1%) |
| Flatulence | 11 | 4 | 1 | 2 | 1 | 19 (61.3%) |
| Abdominal spam | 10 | 4 | - | 2 | 1 | 17 (54.8%) |
| Anorexia/Weight loss | 8 | 2 | 1 | 1 | 1 | 13 (41.9%) |
| Nausea/ Vomiting | 10 | 4 | - | - | 1 | 15 (48.4%) |
| Diarrhea | 5 | 2 | - | 1 | - | 8 (25.8%) |
| Constipation | 5 | - | 1 | 1 | - | 7 (22.6%) |
| Headache | 4 | 1 | 1 | 1 | 2 | 9 (29.03%) |

**Table 4. Association between *Blastocystis* subtypes and the studied variables.**

| Variable | ST1 n(%) | P | ST2 n(%) | P | ST3 n(%) | P | ST4 n(%) | P | ST5 n(%) | P | MIX STs n(%) | P |
|---|---|---|---|---|---|---|---|---|---|---|---|---|
| | **29** | | **3** | | **13** | | **1** | | **2** | | **9** | |
| | **50.9%** | | **5.3%** | | **22.8%** | | **1.8%** | | **3.5%** | | **15.8%** | |
| **Gender** | | | | | | | | | | | | |
| Male (n = 23; 40.1%) | 12 | 0.872[a] | 1 | 1[b] | 4 | 0.423[a] | - | - | - | - | 6 | 0.137[b] |
| Female (n = 34; 59.6%) | 17 | | 2 | | 9 | | 1 | | 2 | | 3 | |
| **Age** | | | | | | | | | | | | |
| 3–20 (n = 20; 35.1%) | 11 | 0.886[a] | 1 | - | 5 | 0.928[a] | - | - | - | - | 3 | 0.969[a] |
| 21–40 (n = 17; 29.8%) | 8 | | 0 | | 4 | | 1 | | 1 | | 3 | |
| >40 (n = 20; 35.1%) | 10 | | 2 | | 4 | | - | | 1 | | 3 | |
| **Water Supply** | | | | | | | | | | | | |
| Tap water (n = 29; 50.9%) | 15 | 0.753[a] | 2 | 1[b] | 6 | 0.208[a] | 1 | - | - | - | 5 | 1[b] |
| No tap water (n = 28; 49.1%) | 14 | | 1 | | 7 | | - | | 2 | | 4 | |
| **Domestic animals (cat, dog,...)** | | | | | | | | | | | | |
| Presence (n = 14; 24.6%) | 7 | 0.346[a] | 1 | 0.481[b] | 4 | 0.251[b] | - | - | - | - | 2 | 1[b] |
| No presence (n = 43; 75.4%) | 22 | | 2 | | 9 | | 1 | | 2 | | 7 | |
| **Mechanical vector (cockroaches, flies.)** | | | | | | | | | | | | |
| Presence (n = 5; 8.8%) | 4 | 0.352[b] | - | - | 1 | 1[b] | - | - | - | - | - | - |
| No presence (n = 52; 91.2%) | 25 | | 3 | | 12 | | 1 | | 2 | | 9 | |

n = number of participants, a: Chi-squared test; b: Fisher test

*statistically significant at p≤0.05

and characterize *Blastocystis* sp. subtypes in 60 individuals from the southern region of Syria suffering from various gastrointestinal symptoms. The infection of *Blastocystis* sp. was detected by direct examination and conventional PCR methods. Our results revealed a high prevalence of infection, reaching 100% using the molecular tool. It was found that the rate of infection varies widely between different countries and between regions within the same country [28–31]; for example, in countries such as Lebanon, Qatar, United Arab Emirates, Saudi Arabia and Libya, the incidence of *Blastocystis* sp. ranged from 44.4% up to 86.6% [1, 32–35]. Whereas other studies from Iran [29], Tunisia [15], Egypt [36] and Lebanon [37] showed low infection rates (8.1%, 13%, 15.3% and 19% respectively). This difference can be explained according to the prevailing epidemiological situation and different climatic conditions [38].

Our data identified five different *Blastocystis* STs (ST1-ST5) in 57 samples (95%). However, three samples remained undefined (5%). This could be explained by the moderate sensitivity of the primers used in genotyping [39] or could be other different subtypes.

**Table 5. Association between dominant *Blastocystis* subtypes and patients' symptoms.**

| | | ST1 (n = 18) | *P* | ST3 (n = 7) | *P* | Mixed STs (n = 5) | *P* |
|---|---|---|---|---|---|---|---|
| **Clinical Symptoms** | | | | | | | |
| **Abdominal pain** | Yes | 11 | 0.972[a] | 4 | 1[b] | 2 | 0.369[b] |
| | No | 7 | | 3 | | 3 | |
| **Flatulence** | Yes | 11 | 0.285[a] | 4 | 1[b] | 3 | 1[b] |
| | No | 7 | | 3 | | 2 | |
| **Abd. Spasm** | Yes | 10 | 0.490[a] | 4 | 0.703[b] | 3 | 0.668[b] |
| | No | 8 | | 3 | | 2 | |
| **Nausea/Vomiting** | Yes | 10 | 0.123[a] | 4 | 0.427[b] | 1 | 0.391[b] |
| | No | 8 | | 3 | | 4 | |
| **Anorexia/WL** | Yes | 8 | 0.214[a] | 2 | 1[b] | 2 | 1[b] |
| | No | 10 | | 5 | | 3 | |
| **Headache** | Yes | 5 | 1[b] | 1 | 0.662[b] | 3 | 0.098[b] |
| | No | 13 | | 6 | | 2 | |
| **Diarrhea** | Yes | 5 | 0.488[b] | 2 | 0.635[b] | 1 | 1[b] |
| | No | 13 | | 5 | | 4 | |
| **Constipation** | Yes | 6 | 0.164[b] | - | - | 1 | 1[b] |
| | No | 12 | | 7 | | 4 | |

n = number of participants; WL: weight lost, a: Chi-squared test; b: Fisher test; *statistically significant at p≤0.05

Previous studies have shown that ST1 and ST3 are the most prevalent subtypes distributed among human individuals and several kinds of animals worldwide [31, 40, 41]. Earlier reports from the Middle East revealed that the most dominant *Blastocystis* subtype was ST3 followed by ST1 [26, 36, 42, 43]. However, in our study, ST1 was the most predominant (66.7%). This finding is in agreement with previous studies conducted in Iran [2, 44], Libya [35], Turkey [45] and the United Arab Emirates [33].

Interestingly, our present study has identified mixed different STs infection in approximately one-sixth of the samples. Mixed STs infection has been reported previously with varying prevalence [8, 46–48] and can be explicated with probable exposure to different sources of contamination [49].

Earlier reports pointed out to a possible high risk of *Blastocystis* sp. infection in people with close contact to animals, supporting the hypothesis of transmission from animals to humans [50, 51]. For example, Ruaux *et al*. found that ST1 was the main subtype amongst patients working in shelter-resident dogs and cats, emphasizing the potential role of domestic animals in human *Blastocystis* sp. infection, while ST3 was more limited to humans [52]. However, in our study, none of the STs identified were significantly associated with the presence of domestic animals or the mechanical vectors either, which may be due to the limited number of the studied samples.

Additionally, in this study, no significant correlation was found between the different drinking water sources and between any *Blastocystis* STs infection. This finding can be explained by the fact that most of our patients stated that they drink tap water, which is generally purified before being distributed to the residences and is in accordance with several previous studies which detected a negative association between *Blastocystis* STs and tap water as a drinking source [19, 53].

The pathogenic role of *Blastocystis* sp. in causing clinical symptoms remains a controversial issue due to its genetic diversity, inability to exclude any other intestinal parasites, bacteria, or

virus causing the same digestive symptoms as well as the different immune responses and the composition of the intestinal microbiota of the host [39, 54, 55].

In this study, 56.9% of individuals infected with ST1 had various gastrointestinal symptoms; however, no significant correlation was found between the different symptoms. This finding disagrees with some previous studies that suggest that *Blastocystis* sp. ST1 may have a pathogenic effect and cause symptomatic infections [42, 56, 57] and is in line with previous data that did not find any statistically significant correlation between *Blastocystis* sp. infection and any gastrointestinal symptoms. It is worth mentioning that half of the samples ~55% in this study were infected with the different STs detected alone, while co-infection with other intestinal parasites were found in 45% of the samples, which leaves the question about the possible real pathogenic potential of this parasite still open [19, 58].

Furthermore, our data showed that females had a higher infection rate compared to males. Yet, no significant association was found between gender and *Blastocystis* infection ($P = 0.872$). This finding is consistent with several previous studies conducted in many countries [35, 59]. Moreover, the incidence of *Blastocystis* infection was relatively similar in all age groups. No association was detected with any particular ST. Our data comes in line with the study of Khaled *et al*. conducted on Syrian refugees living in North Lebanon [49], and with research from Qatar by Abu-Madi *et al*. where no difference was found between the prevalence of *Blastocystis* and between age groups [32]. However, our finding did not support previous reports indicating an association between age and infection being more common in young or adult patients [29, 36].

## Conclusion

This study is the first report from the southern region of Syria to characterize and detect the prevalence of different *Blastocystis* sp. STs. Despite the limited number of the samples, this study showed the predominant distribution of ST1. However, further studies are needed on large samples (humans and animals) and in the different regions of Syria that may lead to a better knowledge of the *Blastocystis* sp. prevalence, transmission sources to humans, interaction with the host, and pathogenicity.

## Acknowledgments

The authors gratefully acknowledge all the patients and their families for taking part in this study. We also thank Ms. Marah Marrawi for her assistance with statistical analyses and Dr Rasheed Abdul Hadi for proofreading.

## Author Contributions

**Data curation:** Buthaina Darwish, Ghalia Aboualchamat, Samar Al Nahhas.

**Formal analysis:** Buthaina Darwish, Ghalia Aboualchamat, Samar Al Nahhas.

**Investigation:** Buthaina Darwish, Ghalia Aboualchamat, Samar Al Nahhas.

**Methodology:** Buthaina Darwish, Ghalia Aboualchamat, Samar Al Nahhas.

**Project administration:** Buthaina Darwish, Ghalia Aboualchamat, Samar Al Nahhas.

**Validation:** Buthaina Darwish, Ghalia Aboualchamat, Samar Al Nahhas.

**Writing – original draft:** Buthaina Darwish, Ghalia Aboualchamat, Samar Al Nahhas.

**Writing – review & editing:** Buthaina Darwish, Ghalia Aboualchamat, Samar Al Nahhas.

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
