## [Decision Letter · Decision Letter 0]

29 Nov 2022

PONE-D-22-27237Molecular characterization of Blastocystis subtypes in symptomatic patients from the southern region of SyriaPLOS ONE

Dear Dr. Nahhas,

Thank you for submitting your manuscript to PLOS ONE. After careful consideration, we feel that it has merit but does not fully meet PLOS ONE’s publication criteria as it currently stands. Therefore, we invite you to submit a revised version of the manuscript that addresses the points raised during the review process.

After a lengthy review, we are pleased to have achieved an adequate assessment of your manuscript.

Note that we had to resort to a third reviewer who has suggested significant changes; I hope you will consider all the indications and improve the validity and quality of your manuscript.

We look forward to receiving your revised manuscript.

Kind regards,

Kovy Arteaga-Livias

Academic Editor

PLOS ONE

Journal Requirements:

Reviewers' comments:

Reviewer's Responses to Questions

**Comments to the Author**

1. Is the manuscript technically sound, and do the data support the conclusions?

Reviewer #1: No

Reviewer #2: Partly

Reviewer #3: Yes

2. Has the statistical analysis been performed appropriately and rigorously? 

Reviewer #1: No

Reviewer #2: Yes

Reviewer #3: No

3. Have the authors made all data underlying the findings in their manuscript fully available?

Reviewer #1: No

Reviewer #2: Yes

Reviewer #3: Yes

4. Is the manuscript presented in an intelligible fashion and written in standard English?

Reviewer #1: No

Reviewer #2: No

Reviewer #3: No

5. Review Comments to the Author

Reviewer #1: Comments to the Authors

This manuscript assesses the occurrence and molecular diversity of the Stramenopile of uncertain pathogenicity Blastocystis sp. in outpatients (n=60) of all age groups with gastrointestinal manifestations seeking medical care in southern Syria. Detection of the protist was simultaneously accomplished by conventional microscopy and PCR using subtype-specific primers. An infection/colonization rate of 100% was obtained by PCR. Five different STs (ST1-ST5) were identified. No Sanger sequencing was conducted for subtype confirmation and allele calling. Basic statistical analyses were conducted to ascertain whether epidemiological or clinical variables were associated with an increased risk of harbouring Blastocystis sp. The manuscript is hampered by methodological issues that have compromised the quality of the obtained results and the conclusions reached by the Authors. Writing of the manuscript is poor, with some sections (e.g., discussion) needing more in-depth comments and analyses integrating the information provided here with that already known from previous surveys. The finding of 15% of samples harbouring mixed subtype combinations is interesting, as it demonstrates that mixed infections might be more frequent than initially anticipated in endemic areas. English grammar and editing need polishing. I recommend reviewing by an English native speaker.

Major issues

1. Please note that there are 30 Blastocystis subtypes currently considered as valid, not 17 as the Authors claim in lines 20 and 49. For subtypes ST1-ST22 please see Stensvold and Clark Trends Parasitol. 2020;36:229-232. For the description of ST23-ST26 see Maloney et al. Parasitol Res. 2019;118:575-582. For the description of ST27 and ST28 see Maloney et al. Parasite Epidemiol Control. 2020;9:e00138. For the description of ST29 see Maloney et al. Parasitol Res. 2021;120:2219-2231. For the description of ST30-ST31 see Maloney et al. Microorganisms. 2021;9:1343. For the description of ST32 see Higuera et al. Front Vet Sci. 2021;8:732129. For the description of ST33 and ST34 see Baek et al. Microorganisms 2022;10:1693. Just with this detail the Authors demonstrated very poor awareness of the current Blastocystis field.

2. It is very unclear why the Authors conducted both microscopy and PCR in all faecal samples collected. What is the rationale of using microscopy under these circumstances? In other words, what is the benefit of conducting microscopy over PCR (a method far more sensitive and specific for the detection of Blastocystis) in this study?

3. Sample size is an issue. Sixty samples is a very limited sample panel to extract robust and credible results, particularly from an statistical point of view.

4. Some of the conclusions reached by the Authors are overstated and not supported by the evidence provided.

5. Lines 46-47: Morphological differentiation among Blastocystis subtypes is not difficult. It is impossible. Please amend.

6. Line 57: I am not aware of any campaign specifically devoted to preventing blastocystiasis. If the Authors have this information, please mention it, otherwise, remove this statement.

7. Line 59: please clearly define what a “resident patient” is. Do you mean inpatient? Perhaps outpatient? Please clarify.

8. Line 66: this section is very poorly written and little informative. Please indicate the inclusion/exclusion criteria followed to recruit patients, and define the clinical manifestations considered. Indicate also the exact location where stool samples were collected, and which hospital/medical centres participated in the study and analysed the samples. Also, provide detailed information on the epidemiological (e.g, which types of drinking water sources were considered? Which domestic animals were considered?) and clinical variables gathered to assess the risk of infection/colonization by Blastocystis sp. Please be as thorough as possible in your description of this section.

9. Line 77: Blastocystis presents vacuolar, avacuolar, granular, amoeboid, and cyst forms. What was the rationale for considering vacuolar forms only for diagnosing the presence of the protist? Why the Authors stablished a cut-off of five vacuolar forms for considering a result positive? Any reference to back up this decision?

10. Line 83: The QIAamp DNA Stool Mini Kit is optimized for using 180-220 mg of faecal sample. Why did the authors decided to use 250-300 mg? By doing this the authors are hampering the performance of the kit to obtain good quality/quantity of genomic DNA.

11. Why did the Authors not submit PCR amplicons of the expected size for Sanger sequencing? This would allow subtype confirmation and probably allele calling. Also submitting representative sequences to GenBank as reference sequences obtained in Syria. All these would have added value to the manuscript.

12. Lines 133 and 140: Current Figures 1 and 2 are not essential and can be moved to the Supplmentary material section or even removed from the text.

13. Discussion section. Please describe what the current know epidemiological situation of Blastocystis in Syria and in neighbouring countries is. See for instance Al Nahhas and Aboualchamat Food Waterborne Parasitol. 2020;21:e00090. Clearly mention what is, in the opinion of the Authors, the main contribution of the manuscript to the field. Importantly, please devote a specific paragraph in the discussion section to list the limitations of the study and how these drawbacks have likely influenced the results obtained and the conclusions reached. For instance, how do the Authors feel about the robustness of their statistical analyses over 60 samples only?

14. Discussion section lines 173-181: please focus the discussion on the epidemiological scenario present in Middle East countries. It makes little sense include information from regions and countries (Australia, Peru, China) where the epidemiology of Blasotcystis can be very different and difficult to compare with. Amend.

15. Discussion section, lines 182-183: this statement is wrong. Please note that ST3 has been widely found in wildlife and livestock. See for instance Hublin et al. Res Vet Sci. 2021;135:260-282. Please make statements based on solid scientific evidence only

16. Discussion section, lines 188-190: any chance of analysing these isolates by Sanger sequencing and determine the subtypes involved?

17. Lines 193-196: please note that in Table 4 no associations were found between a given Blastocystis ST and the occurrence of gastrointestinal disorders. Therefore, this paragraph is overstated and misleading. Please amend and avoid extracting conclusions not supported by the results obtained in the study.

18. Lines 199-205: following my line of reasoning above, and in the absence of epidemiological and molecular data on Blastocystis in domestic animals and wildlife, the Authors cannot infer any zoonotic potential. Again, please avoid overstating and extracting conclusions not supported by the results obtained in the study.

Minor issues

1. Line 14: please clearly indicate which authors contributed equally to the work.

2. Lines 24 and 30: the abstract should include the nature of the variables included in the statistical analyses.

3. Line 26: the abstract should make clear that all 60 faecal samples were simultaneously investigated by microscopy and PCR.

4. Line 41: “sp.” should not be italicised.

5. Line 73: please clearly indicate here how long elapsed between sample collection and DNA extraction and purification.

6. Line 93: 94ºC.

7. Lines 94 and 95: 72ºC.

8. Line 118: please provide the median value instead the mean value. The former is far more informative.

9. Line 118: please see my comments above regarding proper description of clinical variables.

10. Line 169: the meaning of this sentence is apparently contradictory. Please amend.

Reviewer #2: Darwish et al., investigated the prevalence of Blastocystis sp., subtypes in southern regions of Syria. Data about the protist is rare in that region and such studies could be interesting. The manuscript need English writing revision and some clarification before consideration for publishing in the journal.

Abstract

Line: 20-21, the number of subtypes are now 22. Please check the literature (10.1016/j.pt.2019.12.009) and revise it here and through the text.

Introduction

Line 42: … the presence of Blastocystis sp.; however, …

Line 43: after references ([3, 4]) please insert a dot and insert new sentence.

Lines 41-45 need writing revision.

Line 49: please correct as mentioned for abstract. (Now, 22 subtypes have been characterized).

Line 53: rare in humans, but…

You may add some data about the prevalence of Blastocystis in Syria or neighbor countries.

Materials and Methods

Line 64: please terminate the sentence with a dot. … all patients participated in…

Please unify the term “Blastocystis” to “Blastocystis sp.” throughout the text.

Results

Line 127: … isolates.

Line 130: please delete “only”.

L137: mixed subtypes were …

L146: since all 60 samples were positive for Blastocystis with the SSU rRNA gene primers, you cannot rule out the symptoms due to other parasites. Therefore, please delete this sentence and modify your statement. You can discuss this in the discussion.

Discussion

This part is just a comparison between your data and data released by other countries. You should add more discussion about HOW and WHY you obtained this data. You should discuss about the distribution of subtypes in your study, the subtyping methods and translated subtypes from STS primer subtyping (what is discussed in your study) to consensus subtypes, which is mentioned in the “ Terminology for Blastocystis subtypes-a consensus (table 2)”.

Reviewer #3: This manuscript describes the occurrence and distribution of different Blastocystis subtypes detected in 48 samples out of 60 samples of symptomatic patients from southern region of Syria.

The distribution was compared according to age, gender, presence of domestic animals and availability of water supply. Moreover, association of different subtypes (single or mixed) with clinical symptoms was investigated. Five subtypes were identified; with ST1 was the predominant subtype followed by ST3.

Overall, the manuscript is straightforward; the findings are interesting and provide baseline molecular information on the distribution of Blastocystis subtypes in Syria. However, major drawbacks that clearly appear in the study design, data analysis and results presentation should be adequately addressed before this manuscript can be acceptable for publication. Please refer to the comments given below.

MAJOR REVISION

1. Justification of the study should be strengthened. If this is the first study to provide molecular information about Blastocystis in Syria; then this should be clearly stated as a justification; in addition to some epidemiological characteristics of the study area or some population-specific characteristics. If previous studies from other regions of Syria are available then related findings should be presented and research gaps should be stated.

2. The manuscript lacks information about the study design. Information about study design should be added to abstract and methods section. A STROBE checklist, used for the reporting of observational studies, can be downloaded (http://www.strobe-statement.org/index.php?id=available-checklists), filled in applicable manuscript section, and then the completed checklist can be uploaded as a Supporting Information file.

3. The manuscript lacks important information about the study area. What is the name of included city (cities) or districts? Was it a hospital-based or community-based study? What are the characteristics related to intestinal parasites? E.g. sanitation, rural or urban or semi-urban settings, etc. Such information is important for findings generalizability.

4. The manuscript lacks important information about the study population. Information on the recruitment strategy, sample size estimation, sample collection and transport should be provided. Such information is important for findings generalizability.

5. The sample size is small (n = 60). This should be acknowledged as a limitation. Another limitation should also be acknowledged; bacterial and viral causes of symptoms were not ruled out.

6. Data analysis and findings presentation should be improved. Refer to comments # 15-17.

MINOR REVISION

7. Introduction, 2nd sentence: Indeed, pathogenicity of Blastocystis is still controversial. This should be clearly indicated.

8. Line 68: “…from residing patients of the southern regions of Syria…”. What were the included cities or districts or health centers?

9. Line 76: cite a reference for the adopted diagnostic criterion.

10. Line 113: “assess the correlation” change to “assess the association”.

11. Line 122: rephrase the sentence. The word “isolates” can be a source of confusion to readers.

12. Line 124: what about the results of the other 12 patients? Was there any parasites detected or they were negative for any parasites?

13. Line 128: “60 isolates”. Does this refer to the 60 samples or the overall detected subtypes in the 48 samples?

14. Line 137-138: add the percentage after the subtype; i.e. by ST1+2 (5%) and ….

15. Table 2: It can be modified to improve clarity. Remove first total (%) row; and add its data to Single infection row. Then, add a row for co-infection with the data of last row. Then, add a row of Type of co-infection with the 5 rows of combination.

16. Table 4: it should be split into 2 tables. Table 1 for associated factors. Because of small number of positive cases, this table can include all samples; add a column for N examined (n=60) then a column for overall Blastocystis infection, then a column for P. Add columns for ST1 and ST3 with P column for each. The ST1 data should include all ST1 (single and mixed and co-infection; i.e. 34 cases (18+11+5), ST3 should include all ST3 (7+6+1!). Chi square test is not applicable for all variables, indicate where Fischer’s Exact test used. The suggested table design can be found in the attached file of these comments.

17. Table 4: Revise variables grouping. Age groups can be 4 – 20 and the last one should be > 40. Presence of animals should be Presence of domestic animals (Yes or No), why “Presence of insects” is mentioned here?

18. The new table 5 can be for Association of common subtypes and clinical symptoms. Modify the table to avoid repeating Chi square/p value rows. This table should include single infections of related subtypes, as it is. So, add columns for P values after each subtype column. No need to report Chi square values, just the P values and indicate the test used in footnote.

19. Lines 163-165: But these are for ST1 only, as indicated by the p value.

20. Avoid repeating results in discussion.

21. Are there previous findings from other regions of Syria? If yes, compare and discuss. Findings from neighbouring countries (Jordan, Lebanon, Iraq, Turkey, etc) can be used too.

22. Lines 196-198: indicate the limitation here that bacterial and viral causes were not ruled out.

6. PLOS authors have the option to publish the peer review history of their article (what does this mean?). If published, this will include your full peer review and any attached files.

Reviewer #1: **Yes: **David Carmena

Reviewer #2: **Yes: **Hamed Mirjalali

Reviewer #3: **Yes: **Hesham M. Al-Mekhlafi

---

## [Author Response · Author response to Decision Letter 0]

20 Jan 2023

Reviewer #1: thank you for your precise and valuable time and comments, please find below our response to each comment.

Major Issues

1. Please note that there are 30 Blastocystis subtypes currently considered as valid, not 17 as the Authors claim in lines 20 and 49. For subtypes ST1-ST22 please see Stensvold and Clark Trends Parasitol. 2020;36:229-232. For the description of ST23-ST26 see Maloney et al. Parasitol Res. 2019;118:575-582. For the description of ST27 and ST28 see Maloney et al. Parasite Epidemiol Control. 2020;9:e00138. For the description of ST29 see Maloney et al. Parasitol Res. 2021;120:2219-2231. For the description of ST30-ST31 see Maloney et al. Microorganisms. 2021;9:1343. For the description of ST32 see 

2. Higuera et al. Front Vet Sci. 2021;8:732129. For the description ofST33 and ST34 see Baek et al. Microorganisms 2022;10:1693. Just with this detail the Authors demonstrated very poor awareness of the current Blastocystis field. Modification has been done

2. It is very unclear why the Authors conducted both microscopy and PCR in all faecal samples collected. What is the rationale of using microscopy under these circumstances? In other words, what is the benefit of conducting microscopy over PCR (a method far more sensitive and specific for the detection of Blastocystis) in this study? 

Microscopical examination was conducted for all stool samples in order to detect the presence of other types of Parasites (co infection) and to exclude these samples when correlate the possible association with symptoms.

3. Sample size is an issue. Sixty samples is a very limited sample panel to extract robust and credible results, particularly from an statistical point of view. We agree with that point, however previous studies in other countries applied on small/similar number of samples to our study, such as the study Perea et al., 2020 (number of samples is 66); Dogruman et al., 2009 (number of samples is 61)

4. Some of the conclusions reached by the Authors are overstated and not supported by the evidence provided. Thank you, we took this point in consideration. 

Modification has been done

5. Lines 46-47: Morphological differentiation among Blastocystis subtypes is not difficult. It is impossible. Please amend. Modified

6. Line 57: I am not aware of any campaign specifically devoted to preventing blastocystiasis. If the Authors have this information, please mention it, otherwise, remove this statement. Statement has been removed

7. Line 59: please clearly define what a “resident patient” is. Do you mean inpatient? Perhaps outpatient? Please clarify We clarified this point in the M&M and in the Result sections

8. Line 66: this section is very poorly written and little informative. Please indicate the inclusion/exclusion criteria followed to recruit patients, and define the clinical manifestations considered. Indicate also the exact location where stool samples were collected, and which hospital/medical centres participated in the study and analysed the samples. Also, provide detailed information on the epidemiological (e.g, which types of drinking water sources were considered? Which domestic animals were considered?) and clinical variables gathered to assess the risk of infection/colonization by Blastocystis sp. Please be as thorough as possible in your description of this section. This section has been modified taking in consideration all your comments

9. Line 77: Blastocystis presents vacuolar, avacuolar, granular, amoeboid, and cyst forms. What was the rationale for considering vacuolar forms only for diagnosing the presence of the protist? Why the Authors stablished a cut-off of five vacuolar forms for considering a result positive? Any reference to back up this decision? We considered the 5 and plus vacuolar forms as being more accurate to detect, depending on previous reference (added in the text)

10. Line 83: The QIAamp DNA Stool Mini Kit is optimized for using 180-220 mg of faecal sample. Why did the authors decided to use 250-300 mg? By doing this the authors are hampering the performance of the kit to obtain good quality/quantity of genomic DNA. 

This protocol is recommended by the kit itself when the target DNA is not distributed homogeneously in the stool and /or low concentration

11. Why did the Authors not submit PCR amplicons of the expected size for Sanger sequencing? This would allow subtype confirmation and probably allele calling. Also submitting representative sequences to GenBank as reference sequences obtained in Syria. All these would have added value to the manuscript. Thank you. We indeed planning to do so, but unfortunately due to the high coast and the sanctions on Syria this may take a while to accomplish

12. Lines 133 and 140: Current Figures 1 and 2 are not essential and can be moved to the Supplmentary material section or even removed from the text. Thank you for your comment. We removed the figures.

13. Discussion section. Please describe what the current know epidemiological situation of Blastocystis in Syria and in neighbouring countries is. See for instance Al Nahhas and Aboualchamat Food Waterborne Parasitol. 2020;21:e00090. Clearly mention what is, in the opinion of the Authors, the main contribution of the manuscript to the field. Importantly, please devote a specific paragraph in the discussion section to list the limitations of the study and how these drawbacks have likely influenced the results obtained and the conclusions reached. For instance, how do the Authors feel about the robustness of their statistical analyses over 60 samples only? We should clarify that this is the first molecular study on the Blastocystis sp. subtypes in the southern region of Syria.

We took your consideration and made the suitable modification

14. Discussion section lines 173-181: please focus the discussion on the epidemiological scenario present in Middle East countries. It makes little sense include information from regions and countries (Australia, Peru, China) where the epidemiology of Blasotcystis can be very different and difficult to compare with. Amend Modification has been made

15. Discussion section, lines 182-183: this statement is wrong. Please note that ST3 has been widely found in wildlife and livestock. See for instance Hublin et al. Res Vet Sci. 2021;135:260-282. Please make statements based on solid scientific evidence only 

Modification has been made

16. Discussion section, lines 188-190: any chance of analysing these isolates by Sanger sequencing and determine the subtypes involved? We will try to do so as we mentioned above 

17. Lines 193-196: please note that in Table 4 no associations were found between a given Blastocystis ST and the occurrence of gastrointestinal disorders. Therefore, this paragraph is overstated and misleading. Please amend and avoid extracting conclusions not supported by the results obtained in the study. Modification has been done

18. Lines 199-205: following my line of reasoning above, and in the absence of epidemiological and molecular data on Blastocystis in domestic animals and wildlife, the Authors cannot infer any zoonotic potential. Again, please avoid overstating and extracting conclusions not supported by the results obtained in the study. Modification has been done

Minor issues

1. Line 14: please clearly indicate which authors contributed equally to the work. Clarifying has been done

2. Lines 24 and 30: the abstract should include the nature of the variables included in the statistical analyses 

Done

3. Line 26: the abstract should make clear that all 60 faecal samples were simultaneously investigated by microscopy and PCR. Done

4. Line 41: “sp.” should not be italicised.

 Done

5. Line 73: please clearly indicate here how long elapsed between sample collection and DNA extraction and purification. Each stool specimen was divided into two parts: one part was fixed in formalin solution 10% (1:3) for microscopic investigation and the other was stored at -20◦C for molecular studies

6. Line 93: 94ºC. Done

7. Lines 94 and 95: 72ºC. Done

8. Line 118: please provide the median value instead the mean value. The former is far more informative. Done

9. Line 118: please see my comments above regarding proper description of clinical variables. The clinical symptoms were clearly stated in table 3

10. Line 169: the meaning of this sentence is apparently contradictory. Please amend. Clarification has been done

Reviewer #2: thank you for your valuable comments, please find below our response to each comment

Line: 20-21, the number of subtypes are now 22. Please check the literature (10.1016/j.pt.2019.12.009) and revise it here and through the text. Done

Line 42: … the presence of Blastocystis sp.; however, … Modified

Line 43: after references ([3, 4]) please insert a dot and insert new sentence. Modified

Lines 41-45 need writing revision. Modified

Line 49: please correct as mentioned for abstract. (Now, 22 subtypes have been characterized). Done

Line 53: rare in humans, but…

You may add some data about the prevalence of Blastocystis in Syria or neighbor countries. To our knowledge no previous molecular studies were done in other regions of Syria. However, we compared our results with neighboring countries

Line 64: please terminate the sentence with a dot. … all patients participated in…

Please unify the term “Blastocystis” to “Blastocystis sp.” throughout the text. Done

Line 127: … isolates. Done

Line 130: please delete “only”. Modified

L137: mixed subtypes were … Done

L146: since all 60 samples were positive for Blastocystis with the SSU rRNA gene primers, you cannot rule out the symptoms due to other parasites. Therefore, please delete this sentence and modify your statement. You can discuss this in the discussion. We made some modification to the paragraph making it clearer (table 3)

Discussion

This part is just a comparison between your data and data released by other countries. You should add more discussion about HOW and WHY you obtained this data. You should discuss about the distribution of subtypes in your study, the subtyping methods and translated subtypes from STS primer subtyping (what is discussed in your study) to consensus subtypes, which is mentioned in the “ Terminology for Blastocystis subtypes-a consensus (table 2)”. Modifying has been made to the Discussion section

Reviewer #3: thank you for your comments, please find below our response to each comment

Major revision

1.Justification of the study should be strengthened. If this is the first study to provide molecular information about Blastocystis in Syria; then this should be clearly stated as a justification; in addition to some epidemiological characteristics of the study area or some population-specific characteristics. If previous studies from other regions of Syria are available then related findings should be presented and research gaps should be stated This study provided the first genetic characterization of Blastocystis subtypes prevalence in patients from the southern region of Syria (Abstract- Discussion).

2.The manuscript lacks information about the study design. Information about study design should be added to abstract and methods section. A STROBE checklist, used for the reporting of observational studies, can be downloaded (http://www.strobe-statement.org/index.php?id=available-checklists), filled in applicable manuscript section, and then the completed checklist can be uploaded as a Supporting Information file. Study design done 

3. The manuscript lacks important information about the study area. What is the name of included city (cities) or districts? Was it a hospital-based or community-based study? What are the characteristics related to intestinal parasites? E.g. sanitation, rural or urban or semi-urban settings, etc. Such information is important for findings generalizability. Study areas done (study design). The study was based on hospital and health centers samples (results). This study was conducted on the semi urban areas (Sampling collection and the studied specimen) 

4. The manuscript lacks important information about the study population. Information on the recruitment strategy, sample size estimation, sample collection and transport should be provided. Such information is important for findings generalizability. Done (Sampling collection and the studied specimen) 

5. The sample size is small (n = 60). This should be acknowledged as a limitation. Another limitation should also be acknowledged; bacterial and viral causes of symptoms were not ruled out. Done (Discussion and conclusion)

6. Data analysis and findings presentation should be improved. Refer to comments # 15-17. Done

Minor Revision

7.Introduction, 2nd sentence: Indeed, pathogenicity of Blastocystis is still controversial. This should be clearly indicated. Done

8. Line 68: “…from residing patients of the southern regions of Syria…”. What were the included cities or districts or health centers? Done

9. Line 76: cite a reference for the adopted diagnostic criterion. Done

10. Line 113: “assess the correlation” change to “assess the association”. Done

11. Line 122: rephrase the sentence. The word “isolates” can be a source of confusion to readers. Changed to samples

12. Line 124: what about the results of the other 12 patients? Was there any parasites detected or they were negative for any parasites? 14 samples (not 12) out of 60 contained other parasites (table 2) but they were negative for Blastocystis sp. using microscopic examination

13. Line 128: “60 isolates”. Does this refer to the 60 samples or the overall detected subtypes in the 48 samples? We clarified this point in the result section (lines-119-123)

14. Line 137-138: add the percentage after the subtype; i.e. by ST1+2 (5%) and…. Done

15. Table 2: It can be modified to improve clarity. Remove first total (%) row; and add its data to Single infection row. Then, add a row for co-infection with the data of last row. Then, add a row of Type of co-infection with the 5 rows of combination. Done

16. Table 4: it should be split into 2 tables. Table 1 for associated factors. Because of small number of positive cases, this table can include all samples; add a column for N examined (n=60) then a column for overall Blastocystis infection, then a column for P. Add columns for ST1 and ST3 with P column for each. The ST1 data should include all ST1 (single and mixed and co-infection; i.e. 34 cases (18+11+5), ST3 should include all ST3 (7+6+1!). Chi square test is not applicable for all variables, indicate where Fischer’s Exact test used. The suggested table design can be found in the attached file of these comments. Done

17. Table 4: Revise variables grouping. Age groups can be 4– 20 and the last one should be > 40. Presence of animals should be Presence of domestic animals (Yes or No), why “Presence of insects” is mentioned here? Done

Insects may play a mechanical role in the transmission of Enteric Parasites (Patel et al., 2022)

18. The new table 5 can be for Association of common subtypes and clinical symptoms. Modify the table to avoid repeating Chi square/p value rows. This table should include single infections of related subtypes, as it is. So, add columns for P values after each subtype column. No need to report Chi square values, just the P values and indicate the test used in footnote. Done

19. Lines 163-165: But these are for ST1 only, as indicated by the p value. Modified/Done

20. Avoid repeating results in discussion Done

21. Are there previous findings from other regions of Syria? If yes, compare and discuss. Findings from neighbouring countries (Jordan, Lebanon, Iraq, Turkey, etc) can be used too. To our knowledge no previous molecular studies were done in other regions of Syria. However, we compared our results with neighboring countries

22. Lines 196-198: indicate the limitation here that bacterial and viral causes were not ruled out. Done (discussion and conclusion)

---

## [Decision Letter · Decision Letter 1]

21 Feb 2023

PONE-D-22-27237R1Molecular characterization of Blastocystis subtypes in symptomatic patients from the southern region of SyriaPLOS ONE

Dear Dr. Nahhas,

Thank you for submitting your manuscript to PLOS ONE. After careful consideration, we feel that it has merit but does not fully meet PLOS ONE’s publication criteria as it currently stands. Therefore, we invite you to submit a revised version of the manuscript that addresses the points raised during the review process.

We look forward to receiving your revised manuscript.

Kind regards,

Kovy Arteaga-Livias

Academic Editor

PLOS ONE

Journal Requirements:

Reviewers' comments:

Reviewer's Responses to Questions

**Comments to the Author**

1. If the authors have adequately addressed your comments raised in a previous round of review and you feel that this manuscript is now acceptable for publication, you may indicate that here to bypass the “Comments to the Author” section, enter your conflict of interest statement in the “Confidential to Editor” section, and submit your "Accept" recommendation.

Reviewer #1: (No Response)

Reviewer #3: All comments have been addressed

2. Is the manuscript technically sound, and do the data support the conclusions?

Reviewer #1: Partly

Reviewer #3: Yes

3. Has the statistical analysis been performed appropriately and rigorously? 

Reviewer #1: No

Reviewer #3: Yes

4. Have the authors made all data underlying the findings in their manuscript fully available?

Reviewer #1: Yes

Reviewer #3: Yes

5. Is the manuscript presented in an intelligible fashion and written in standard English?

Reviewer #1: Yes

Reviewer #3: Yes

6. Review Comments to the Author

Reviewer #1: Minor issues

1. Line 30: rRNA not italicised. Amend here and through the whole manuscript.

2. Line 31: 100% (without space).

3. Line 50: Please note that four new Blastocystis STs (ST35-ST38) have been recently described. See Maloney et al. Microorganisms. 2022;11(1):46. doi: 10.3390/microorganisms11010046. Please update this information.

4. Line 68: In my previous appraisal, I specifically requested improving this subsection by providing information on inclusion/exclusion criteria followed to recruit patients, defining the clinical manifestations considered, and indicating the hospital/medical centres participating in the study. These requests, which are still pending.

Reviewer #3: The authors have adequately addressed all my comments highlighted on the first version of the manuscript.

7. PLOS authors have the option to publish the peer review history of their article (what does this mean?). If published, this will include your full peer review and any attached files.

Reviewer #1: **Yes: **David Carmena

Reviewer #3: **Yes: **Hesham M. Al-Mekhlafi

---

## [Author Response · Author response to Decision Letter 1]

27 Feb 2023

Reviewer #1: 

We would like to thank you for your precise and valuable time and comments, please find below our response to each comment.

1. Line 30: rRNA not italicised. Amend here and through the whole manuscript.

Done

2- Line 31: 100% (without space).

Done 

3. Line 50: Please note that four new Blastocystis STs (ST35-ST38) have been recently described. See Maloney et al. Microorganisms. 2022;11(1):46. doi: 10.3390/microorganisms11010046. Please update this information. 

Modification has been done

4. Line 68: In my previous appraisal, I specifically requested improving this subsection by providing information on inclusion/exclusion criteria followed to recruit patients, defining the clinical manifestations considered, and indicating the hospital/medical centers participating in the study. These requests, which are still pending

We did not have any exclusion criteria; we only included all patients suffering from different gastrointestinal symptoms. 

We modified all other requested points.

---

## [Editor Report · Decision Letter 2]

6 Mar 2023

Molecular characterization of Blastocystis subtypes in symptomatic patients from the southern region of Syria

PONE-D-22-27237R2

Dear Dr. Nahhas,

We’re pleased to inform you that your manuscript has been judged scientifically suitable for publication and will be formally accepted for publication once it meets all outstanding technical requirements.

Kind regards,

Kovy Arteaga-Livias

Academic Editor

PLOS ONE
---

## [Editor Report · Acceptance letter]

9 Mar 2023

PONE-D-22-27237R2 

Molecular characterization of *Blastocystis* subtypes in symptomatic patients from the southern region of Syria 

Dear Dr. Al Nahhas:

I'm pleased to inform you that your manuscript has been deemed suitable for publication in PLOS ONE. Congratulations! Your manuscript is now with our production department. 

Kind regards, 

on behalf of

Dr. Kovy Arteaga-Livias 

Academic Editor

PLOS ONE